# Primary Healthcare Nurse’s Barriers and Facilitators to Providing Sexual and Reproductive Healthcare Services of LGBTQI Individuals: A Qualitative Study

**DOI:** 10.3390/healthcare10112208

**Published:** 2022-11-03

**Authors:** Raikane James Seretlo, Mathildah Mpata Mokgatle

**Affiliations:** Department of Public Health, Sefako Makgatho Health Sciences University, Ga-Rankuwa 0208, South Africa

**Keywords:** sexual and reproductive healthcare services, LGBTQI, primary healthcare, nurses, qualitative study

## Abstract

In most cases, we only hear Lesbians, Gays, Bisexuals, Transgender, Queer, and Intersex (LGBTQI) patients complaining about nurses being the reason for not accessing and utilizing healthcare services; for example, studies reports on the different attitudes of healthcare providers including nurses against LGBTQI patients. However, factors influencing the behavior of South African Primary Healthcare (PHC) Nurses toward LGBTQI patients are rarely reported. The study aimed to explore how PHC nurses experienced and perceived sexual and reproductive health services for LGBTQI individuals in Tshwane, Gauteng Province, South Africa. The study followed qualitative research using an exploratory design approach. The sample included 27 PHC nurses from Tshwane, Gauteng Province, South Africa. In-depth face-to-face interviews were coded and analyzed using Thematic Content Analysis (TCA) which included five interrelated steps. The results revealed three main themes: barriers to the provision of LGBTQI-related SRHS, facilitators for the provision of SRHS to LGBTQI individuals, and strategies to improve LGBTQI individuals’ SRHS accessibility and availability. Common barriers were related to the institutions, PHC nurses, the general public, and LGBTQI patients themselves. Regardless of the challenges faced by PHC nurses, there were some enabling factors that pushed them to continue rendering SHRS to LGBTQI patients who came to their clinics. Almost all PHC nurses suggested the importance of awareness, transparency, collaboration, and the need for training related to LGBTQI healthcare issues.

## 1. Introduction

Primary Health Care (PHC) is progressing fairly in South Africa regardless of the backdrop of the significant health system and social and economic challenges [1]. PHC Nurses are considered the first point of entry to the healthcare facilities for all types of patients to provide different healthcare services, including sexual and reproductive healthcare services (SRHS). However, generally, nurses in South Africa are faced with skills, knowledge, and training challenges when providing SRHS to patients [2]. Specifically, studies conducted overseas have shown that nurses lack foundational knowledge and expertise in LGBTQI health needs [3,4,5,6]. This is because nurses were not taught during their studies about LGBTQI health needs [7].

Currently, in South Africa, studies have shown that a lack of resources, unsupported structural environments, and lack of policies and guidelines for LGBTQI patients in the healthcare facilities are some of the hindering factors for PHC nurses to continue providing SRHS to LGBTQI patients [8,9,10,11]. Moreover, the literature reports that PHC nurses experience difficulties in providing SHRS to LGBTQI individuals; this is due to unsupported institutional conditions and a lack of training, skills, and knowledge about LGBTQI health-related issues [7,12,13]. Additionally, PHC nurses stipulated that their cultures and religious beliefs were some of the reasons that caused difficulties in interacting with LGBTQI patients [7,12,13].

Most results from different studies indicate discrimination, judgmental, prejudicial, bad labeling comments, treatment of LGBTQI individuals by nurses, reproductive health issues such as pregnancy-related challenges, and lack of fertility support [14,15,16,17].

Authors observed that South African nurses experience the same challenges SRHS of lack of training, skills, resources, limited knowledge, and inclusivity for LGBTQI. Therefore, if the above-mentioned are not addressed, nurses will continue with uncertainty and fear while rendering care [18], resulting in poor care services and leading to increased numbers of sexually transmitted infections (particularly HIV/AIDS) [19]. The heteronormative approach amongst nurses will continue [3,12] because they will always treat LGBTQI individuals with an assumption of male and identify as female without dealing with their SRHS-specific needs such as lubricants, contraceptives, and hormonal treatment; as a result, leaving LGBTQI individuals faced with an increased danger of indirect and direct consequences of potentially traumatic events (PTEs) vulnerability, such as hate crimes and childhood abuse [20]. Nurses will continue expressing stereotypical beliefs about LGBTQI individuals, thus creating an uncomfortable climate for LGBTQI individuals [18,21].

Little is known about South African PHC nurses’ barriers and facilitators during the provision of SHRS for LGBTQI individuals in South Africa. Therefore, the aim of this article is to explore barriers and facilitators perceived and experienced by PHC nurses during the provision of SRHS around Tshwane, Gauteng Province of South Africa, in order to document challenges experienced by PHC nurses to provide recommendations to the policymakers in amending, adding, and developing improved healthcare protocols that would assist in LGBTQI inclusivity in the primary health facilities and the healthcare system at large.

## 2. Materials and Methods

### 2.1. Type of Study

The study used a qualitative design to acquire in-depth experiences and perceptions of primary healthcare nurses who are providing sexual and reproductive health services for LGBTQI individuals. An exploratory approach guided by a semi-structured interview guide and face-to-face interviews was used. After interviewing 24 primary health care nurses who were the participants of the study, data saturation was reached. The researchers stopped interviews at the 27th interview. Data saturation means the stage at which no new information is produced during the data collection process [22]. The conducted interviews lasted between 25–45 min and were transcribed verbatim, and NVivo 12 software was used to analyze them.

### 2.2. Population and Sample Size

The researchers included primary healthcare nurses of different ages and gender from the 8 primary healthcare facilities in the study to represent the wider possible range of experiences and perceptions.

Appointments were arranged with various and relevant facilities’ managers over the phone; in other healthcare facilities, the researcher personally visited the study settings to make appointments for interviews. On data collection day, participants were recruited within their primary healthcare facilities during working hours. Mostly, recruitment occurred during staff morning meetings. The researcher explained the purpose and objectives of the study to all primary health care nurses during their morning prayer devotion and meetings. Some participants volunteered immediately after the meeting to participate in the study, and the researcher followed up with other PHC nurses after the meeting during consultation time to ask for their participation. Those who agreed to participate were given an informed consent form to sign. In total, there were 23 who identified as females and 4 who identified as males between the age of 27–63 years who participated in the study.

### 2.3. Study Context

The study was conducted in the City of Tshwane Metropolitan Municipality in the Gauteng Province of South Africa. Tshwane is the capital city of SA. The study was conducted in the region, 1 community health center (CHC), which operates 24 h a day, and 10 clinics that operate from 07:30 until 16:00 every day of the week were selected (The information was obtained from the nurse managers of the selected study settings). The health facilities were selected because they all had professional nurses and these nurses provide and promote sexual and reproductive health care services. The facilities provide comprehensive services, including women’s health (family planning and cancer screenings); child health (immunization); HIV/AIDS/TB and STI; maternal and child health care services (basic antenatal care, intrapartum care); curative services (minor alignment and wound dressings) and chronic disease services (hypertension and diabetes). The city itself has over 700,000 inhabitants, while the much-larger urban area has a population of 2,125,000. The city has different types of religious beliefs, such as Christians, Muslims, Hundi, and Indigenous beliefs. This city is also regulated by one of the South African law act Bill of rights Section 2.7 No of 1996, which states every person is not to be discriminated against regardless of age, gender, race, class, and disability. This act also advises the services rendered by the South African Department of Health.

### 2.4. Interview Guide

A semi-structured interview guide was used to gather data for the study. The interview guide had general (which included general questions about SRHS rendered in the clinics), main (which included specific questions on LGBTQI SRHS), and probing questions. There were open-ended questions in order to gather more rich data on how PHC nurses experienced and perceived SRHS for LGBTQI individuals for LGBTQI individuals. The interview guide was informed, adapted, and formulated based on related literature [15,23,24].

Three participants were used as a pre-test before the main data collection started, and the results were included in the overall data to ensure that the interview guide asked what is supposed to; this was conducted to improve the depth of the data collected. Modification of an interview guide was done accordingly, following the findings from the pre-test study. This included minor changes regarding how questions can be phrased based on the answers from PHC nurses who were in the pilot study, and no questions were removed. An outline of the interview guide is shown below:

**Table d64e265:** 

List of Questions Asked of PHC Nurses for the Qualitative Analysis
Demographic questions: Place of data collection, age, gender, marital status, rank & specialty, and how long have you been a nurse? General questions: What type of services do you render in your institution as prescribed by the Department of Health? Main questions: 2. Main: Could you please share your experiences when rendering SRHS to LGBTQI? Probe: - How do you feel about the current services offered to them? - How was your interaction with LGBTQI patients? - What are your perceptions regarding the quality of sexual and reproductive healthcare services around your health facility? 3. Main: What utmost challenges have you experienced when you have to render SRHS to the LGBTQI individuals in your clinic? Probe: - What makes it difficult? - How prepared/comfortable are you working with LGBTQI patients? (condom use, profoundly oral sex, anal sex, sex toys). 4. Main: What keeps on encouraging you to continue providing SRHS to LGBTQI individuals in spite of the challenges? Probe: - How did/do you learn to provide sexual and reproductive health care services to LGBTQI individuals? (type of training received). 5. Main: What can you suggest so that SRHS for LGBTQI individuals can be improved? Probe: - What other suggestions do you have in mind that you think can help improve sexual and reproductive health care services for LGBTQI individuals?
All interviews were conducted in English, although participants were allowed to respond and express themselves in their own language. Most participants responded and expressed themselves in Setswana, later translated into English by both authors (refer to Section 2.6 for more details).

### 2.5. Data Collection Procedure

Data were collected by using a semi-structured interview guide and face-to-face interviews.

The recruitment of the participants took place after all the approval from the above-mentioned committees was received. The PHC nurses were informed that the digital recorder would be used during the interview session. The audios were transcribed verbatim, anonymized, and analyzed using NVivo 12 software. Furthermore, demographic data that was obtained from the PHC nurses were; place of data collection, gender, age, duration of work as nurses, and any nursing specialty. Twenty-seven interviews were conducted in a private and safe room, and all COVID-19 safety measures were adhered to.

### 2.6. Coding and Data Analysis

The researchers analyzed data using thematic content analysis by Tolley et al., 2016 [25], which included five interrelated steps which included inductive and deductive data process of data coding and thematic framework using NVivo 12 software [25]. The first step was transcribing audio verbatim and translating some of the Setswana audio into English. Both authors are Setswana-speaking. They loaded audio into Ms 350 to translate to English as listening to audio more often without losing the meaning. Secondly, the researchers developed a codebook as an initial coding process while inserting identified themes and subthemes. The third step was thorough reading and examination of data to get meanings from the initial coding folder. In the 4th step, data of the initial coding and application of new codes as they emerged during data analysis were transferred into NVivo 12 software. The raw demographic data were summarized using Microsoft Excel. Interpretation of data was the last step of data analysis, whereby the researchers found more meaningful data and applied the identified main themes and related sub-themes. The process was led by 1 researcher, whereby it was conducted after receiving training from the university, the researcher handed over the analyzed data to the supervisor as a peer reviewer to ensure objectivity and replication.

### 2.7. Ethical Considerations

The study commenced after the approval of the Ethics Committee of Sefako Makgatho Health Science University (SMUREC) protocol number SMUREC/H/203/2021: PG. The research and Ethics Committee of Sefako Makgatho Health Science University (SMUREC) approved the study Protocol number SMUREC/H/203/2021: PG. After ethical approval, the researcher applied for permission to conduct the study from the Tshwane Research Committee, the PHC manager for Tshwane provincial clinics, and the office of skills development for Tshwane municipal clinics.

The participants had the right to withdraw from the study at any time during the research process. Anonymity was maintained throughout data collection and analysis, Participants were requested not to mention their names, and each participant was given a false name, such as P1, which stands for participant 1, and each had a number. The researcher addressed them as P and added a number. Participants were protected, and their safety was ensured by adhering to COVID-19 safety measures.

## 3. Results

Overall, 27 PHC nurses participated in the study. Four (4) of them were men, and 23 were women from eight primary healthcare facilities around Tshwane town. The mean age of the sample was 42 years (SD = 11.39), with a median age of 39 years. Thematic Content analysis for qualitative was used, and three main themes and related sub-themes emerged during data analysis, namely; barriers to the provision of LGBTQI-related SRHS, facilitators for the provision of SRHS to LGBTQI individuals, and strategies to improve LGBTQI individuals’ SRHS accessibility and availability highlighted in Table 1 below: (YOS stands for Years of Service, GN, General nursing, PHC, Primary Health Care Nursing, and ONC, Occupational Nursing).

Theme 1: Barriers to the Provision of LGBTQI-Related SRHS

Institutional-related barriers: PHC nurses specified that most of their clinics are not LGBTQI-user friendly. Furthermore, PHC nurses believe that LGBTQI is afraid of accessing SRHS due to discrimination within their clinics and also not offering appropriate services. Some PHC nurses stipulated that they cannot assist some of the LGBTQI individuals because their clinics are not the ‘right places to deal with sexual and reproductive health-related challenges experienced by LGBTQI individuals.’ The nurse reported that they opted to refer them to “the right clinic” that caters to LGBTQI individuals. According to the PHC nurses, the South African government, specifically the Department of Health, is giving less attention to LGBTQI services and limited and generalized training regarding LGBTQI health-related matters. The nurses further explained that the matters of SHRS for LGBTQI are always included briefly under workshops for vulnerable groups without emphasis, focus, and depth. Lastly, PHC nurses believe that a shortage of formal training, prohibiting and unavailable services, guidelines, policies insufficient of resources, and a lack of government assistance are the major institutional barriers to easily accessible and user-friendly SRHS for LGBTQI individuals. This is illustrated in the quotations below:

“They don’t have services; they are afraid to come because of the discrimination, and another problem is because we do not offer them the proper services”.(Identified as Female, no specialty, 17 YOS)

“I remember in 2020, we had one gay patient who was complaining about anal pain; it looked like after sexual intercourse, he was experiencing some pains and bleeding. We couldn’t assist him and referred him to the other clinic that deals specifically with LGBTQI patients. Yes, we referred him to the right clinic catering to his needs”.(identified as female, No specialty, 11 YOS)

“It’s something that is overlooked. It’s not something that is given attention by the programs that are offered by the Department of Health because I’ve never seen any training or a course specifically on LGBTQI matters. In most cases, when they train us, it will be just generalizing and focused on normal males and identified as females”.(identified as female, PHC specialty, 29 YOS)

PHC nurses-related barriers: PHC nurses’ behaviors, attitudes, lack of training, knowledge, and skills emerged as the factors which made the provision of SRHS to LGBTQI individuals awkward. These included actions such as finding LGBTQI topics difficult to talk about freely, being fearful of hurting LGBTQI individuals during consultations, being scared of stepping into LGBTQI individuals’ personal spaces, and being afraid of offending them.

“I do not think I want to talk to them about their sexual activities; I do not have interest in it that much”.(identified as male, GN specialty, 13 YOS)

“Uh, LGBTQI people are not my topic of interest; I am not ready, always when I see any of them, I just give them what they ask or ignore there at all”.(identified as female, PHC specialty, 20 YOS)

“I am always scared of talking to them; I feel like I will offend them, and I would not render proper skills to them. Yes, I am not skilled, and I am not much interested because of my beliefs. I know I pledged, but this is different”.(identified as female, PHC specialty, 6 YOS)

Social stigma-related barriers: Stigma by the public against LGBTQI individuals was one of the hindrances to accessing and making use of SRHS, as indicated by PHC nurses. PHC nurses believe that some of the reasons LGBTQI individuals are not accessing and utilizing SRHS more or less the same as heterosexual individuals are because of community members’ stigma. The PHC nurses indicated that LGBTQI individuals are scared of being judged against their sexuality and sexual orientation as they visit their clinics. Moreover, the PHC nurses indicated that they are not being judged only by community members but also by patients who are the ques and other staff members who are not nurses.

“I think the biggest problem among these people is stigma. If the stigma can be removed and they can be seen like every individual that comes through the facility”.(identified as female, PHC specialty, 40 YOS)

“I feel it’s very sad that people still have the stigma and that they don’t want to use services because they are scared of being judged”.(identified as female, PHC specialty, 12 YOS)

LGBTQI little openness as a barrier: PHC nurses believe that the challenge of LGBTQI individuals to being open and transparent about their sexuality, sexual practices, and their health complaints are some of the hinders for them to render relevant and related SRHS to them. It was indicated that most LGBTQI people who act straight are not free because of fear of judgment and are more defensive to protect themselves, which ends up leading to misdiagnosis and mistreatment.

“Some are not that open about their sexuality and try to act straight, so they’ll come and present with a different problem. They commonly present with piles. Then when you examine when you assess, obviously you hear from their voice, mannerism”.(identified as female, no specialty, 5 YOS)

“What I’ve observed is, like most of them, they don’t feel free to express themselves, and they fear to be judged”.(identified as female, PHC specialty, 17 YOS)

“They are secretive and mostly because they are used to being judged against, so they come ready to protect themselves, more defensive”.(identified as female, no specialty, 2 YOS)

Theme 2: Facilitators for Provision of SRHS to LGBTQI Individuals

Facilitators are the factors that enabled PHC nurses to continue rendering SRHS to LGBTQI individuals. PHC nurses mentioned a variety of factors that enabled them to continue with the provision of SRHS to LGBTQI individuals in spite of experiencing barriers, as highlighted in Theme 1.

The obligation of governance and compliance: PHC nurses continued rendering services because they had to follow the organizational standards and regulations and function according to their scope of practice in the best way. They believe that their scope of practice and job descriptions keep them pushing regardless of all the challenges. PHC nurses emphasized the importance of making LGBTQI individuals comfortable and providing them with quality health care as part of their job descriptions.

“The thing is, at the end of the day, I have to offer the service, whether I’m comfortable or not comfortable, but at the end of the day, I must provide quality care, yeah”.(identified as female, PHC specialty, 6 YOS)

LGBTQI individuals as facilitators: Most PHC nurses indicated that they continued rendering SRHS to LGBTQI individuals because they are learning from them. They specified that some of the LGBTQI individuals are too kind to share with the PHC nurses about their SRHS needs, including understanding their lifestyle and sexual practices. Moreover, PHC nurses acknowledged that there are a lot of things they do not know that LGBTQI individuals are willing to share, which enriches their understanding.

“I’m also teaching myself; it’s a learning curve. They teach us about things that we didn’t know that they are happening”.(identified as female, PHC specialty, 40 YOS)

“I’m so comfortable because if there’s one thing about them, they will explain to you and make you understand them better”.(identified as female, PHC specialty, 29 YOS)

PHC nurses’ age: Some of the PHC nurses expressed that because of their age, it is, therefore, easier to provide SRHS to LGBTQI people regardless of a lack of training. PHC nurses reported that the younger generation finds it easier to talk and relate with LGBTQI individuals without judging them. Some described that time has changed, and things are no longer the same as in the ancient days when sex was only vaginally or between two different genders.

“I think as the younger generation, it becomes easier to provide sexual and reproductive health care services to them. I can say it’s easier for the younger generations. Yeah, we are able to establish that rapport”.(identified as female, no specialty, 2 YOS)

“I’m young, I’m comfortable, and it’s not like in the olden days where sex was just the normal vaginal. I think some of the things that LGBTQI people do we are exposed to, and we read about them”.(male, no specialty, 1 year of service)

Acquired work experience: Other PHC nurses stipulated that because of their work experiences and knowledge acquired from specific courses, they kept on providing SRHS for LGBTQI individuals.

“I am currently rendering youth-friendly services, so I think that helped me to understand some of the topics because they are covered in the youth-friendly service. They do talk about different sexual orientations, different sexual people, who are not necessarily from LGBTQI, but we render all types of services or anyone who come to our facility”.(identified as female, PHC specialty, 5 YOS)

Existence of LGBTQI acquaintance: Having family members and friends who are LGBTQI has been associated as a helper in rendering SRHS to LGBTQI individuals who visit clinics. PHC nurses see any need to judge people to whom they can relate. As a result, rendering SRHS to LGBTQI individuals is normal, and they treat them like any patient.

“We’ve got friends who are members of the LGBTQI at our homes, so it makes it easier”.(identified as female, no specialty, 2 YOS)

The use of mass media: The PHC nurses reported that different types of mass media, such as Google, the internet, television, newspapers, and social media platforms, are the main facilitators for them to continue offering and providing SRHS to LGBTQI individuals.

“We are self-taught; we use Google or the internet to search for information on how you deal with their challenges or health needs”.(identified as female, no specialty, 11 YOS)

“Nonetheless, whether we like it or not, we hear about it on the TV, read about it, see it on social media and other platforms”.(identified as female, PHC specialty, 13 YOS)

Theme 3: Mastery of Strategies to Improve LGBTQI Individuals’ SRHS

PHC nurses came up with suggestions of how SRHS for LGBTQI individuals can be improved and upgraded to achieve well-being and good health for LGBTQI individuals. In addition, PHC nurses explained how ready they are to collaborate with the government in ensuring that SRHS is easily accessible and more accommodating to LGBTQI individuals.

Awareness and transparency of LGBTQI-focused services: PHC nurses indicated the importance of open access to LGBTQI SRHS. The majority of PHC nurses indicated that there should be a lot of discussions, ongoing engagements, programs, campaigns, and in-service training about LGBTQI SHRS-related issues in public spaces, on radios, and on televisions. PHC nurses hope that these suggested activities will achieve LGBTQI acceptance by society, spread knowledge, and emphasize the importance of inclusivity for LGBTQI individuals. According to the PHC nurses, notice boards should be placed at the entrance of each clinic to show visibility of services that are rendered, and they believe transparency should be done as any other service within clinics.

“I think if we can have a lot of discussions, ongoing engagements, and in-service training, things will get better for us to be able to provide health education to them. LGBTQI people should have their own programs and topics, so that is acceptable and/or well-known”.(identified as male, no specialty, 8 YOS)

“Clinics must indicate notice boards at their gates that indicate that LGBTQI like HIV Pre-exposure prophylaxis (PrEP) and others are rendered”.(identified as male, no specialty, 11 YOS)

Specialization or integration of SRHS: There were a lot of disagreements and suggestions from the PHC nurses; that is, SRHS for LGBTQI should be rendered as a specialized and/or integrated service. Moreover, they suggested that these ideas of isolation should be treated like adolescent youth-friendly services, whereby any LGBTQI individuals are given the highest priority. Whereas some PHC nurses believed that separation and isolation of services would mean they were being stigmatized. Additionally, clarified the importance of integrating SRHS for LGBTQI with other available healthcare services.

“Clinics can make a special department for LGBTQI, whereby people in that department will be trained on how to handle issues pertaining to their needs specifically because someone wants to be transgender, you understand so you know how to advise them better. So, I think it will be best if all the clinics have special services”.(Identified as Female, No specialty, 11 YOS)

“If we can open a clinic and say it is for LGBTQ people, it will be a kind of stigmatizing. So, I think they have to attend all clinics so that they can be seen at any other facilities that are closer to them without being discriminated against or being stigmatized, or secluded in a certain corner or a certain room. They should be able to mix with other patients”.(identified as female, PHC specialty, 25 YOS)

Need for training: All PHC nurses mentioned a need for skills and development to be able to provide relevant services to LGBTQI individuals freely and in peace.

“The training of the staff about the LGBTQI group is important so that we know more about services they require and can better advise them”.(Identified as Female, PHC specialty, 10 YOS)

“Firstly, I think there is a need for training for all health care workers, including security officers and admin working in the facility”.(Identified as Female, PHC specialty, 15 YOS)

## 4. Discussion

The study aimed to explore how PHC nurses experienced and perceived SHRS for LGBTQI individuals in Tshwane, Gauteng Province, South Africa.

In this study, we found three main themes and their related sub-themes. The first theme was about factors that hinder the provision of SRHS by PHC nurses to LGBTQI individuals. The sub-themes of theme 1 included institutional, PHC nurses, social stigma, and LGBTQI little openness-related barriers.

The second theme comprised the most important facilitators for PHC nurses to carry on with giving SRHS to LGBTQI individuals regardless of the challenges experienced. Theme 2’s sub-themes are; the obligation of governance and compliance, LGBTQI individuals as educators, PHC nurses′ work experience and age, the existence of acquaintances, and the use of mass media.

Lastly, the third theme covered ways that might help in improving accessibility, utilization, and availability of SRHS amongst LGBTQI individuals. This theme had three related sub-themes; training for PHC about LGBTQI-related health issues, isolation and collaboration of SRHS, awareness, and transparency of SRHS.

Our study suggests that some of the institutional hindrances like policies, restrictive services, resources shortage, guidelines, and the government’s inability to assist were the reasons that pushed LGBTQI individuals away from accessing SRHS. For example, it was noticed that there are no LGBTQI-specific guidelines and protocols, there is a lack of contraceptive services such as lubricants in the facilities, and SRHS is mostly focused on those who identified as female and male. Moreover, the consequences of LGBTQI clients’ being less open about their sexuality and sexual practices hindered PHC nurses’ in rendering correct and proper SRHS to them. For instance, LGBTQI clients will not report the exact health issues they came up with due to fear of being judged. Again, PHC nurses’ inadequate and poor skills, knowledge, and expertise, and public and societal stigma were some of the barriers to LGBTQI clients accessing SRHS. That is, most PHC nurses do not know how to render SRHS for LGBTQI clients due to a lack of skills, and they indicated that they were not taught during their learning period. All these findings are supported by several studies that cited barriers such as heteronormativity among healthcare providers, judgemental attitudes, limited and restricted youth openness with healthcare providers, fear of disclosing to parents or guardians due to family stigma, and fear of offending LGBTQI patients [26,27]. In addition, another study supported that there are no practice policies and guidelines in LGBTQI healthcare in SA [7]. In support of our findings, studies from overseas revealed that individual factors, social-structural factors, systemic inequities, and inadequate health insurance coverage were some of the barriers for LGBTQI clients to accessing health care services [28,29,30].

Some PHC nurses indicated that their age and the availability of their LGBQTI acquaintances help them to be non-judgemental and always comfortable and available to work with LGBTQI individuals. In particular, young PHC nurses indicated that they fully understand the times they are living in; they also have friends and relatives who are part of the LGBTQI community, and, as a result, it is easier for them to render services to any member of the LGBTQI community. This is in line with another study that indicated that nurse’ experiences and knowledge during care provision for LGBTQI patients for the nurses was moderately influenced and increased by their having LGBTQI friends and relatives [31].

Regardless of the experienced challenges by PHC nurses, the study observed that they were enabled to continue rendering SRHS to LGBTQI clients due to governance and compliance obligations. For example, some PHC nurses indicated that they are rendering these services to LGBTQI clients due to their professional body and scope of practice. This agrees with the study by Dorsen and Van Devanter that indicated that nurse practitioners struggled when caring for LGBQTI patients with their personal and professional values; regardless of their struggle, they had to provide best practices to their patients [24]. Moreover, the use of mass media by PHC nurses was identified as one of the facilitators for gathering information about LGBTQI sexual practices and sexual and reproductive needs. For example, PHC nurses indicated that they learn how to render services and what SRHS needs for LGBTQI through television and social media platforms. Supported, a study by Brown et al. indicated that nurse practitioners and nurse practitioner students’ LGBT Health perceived social media, especially TV, as a major role player in normalizing being LGBTQI by providing a level of acceptance, more familiarity, and more comfortability with the idea of one being LGBTQI [32]. However, other studies have shown different interventions that assisted different healthcare providers in overseas understanding LGBTQI health-related matters. For example, a web-based LGBTQI cultural competency training intervention for oncologists was conducted, and the results showed that training was accepted and improved their skills, knowledge, and attitudes regarding LGBTQI people [33]. Another example is a 10-h LGBTQI curriculum conducted for medical students, and it revealed only self-confidence in working with LGBTQI clients [34].

The majority of the participants displayed an understanding of how SHRS could be improved to accommodate the health and well-being of LGBTQI individuals. This correlates with the findings from several studies that indicated that the negative consequences of a lack of SRHS for LGBTQI individuals were understood by the PHC nurses. For example, increased sexually transmitted illnesses like HIV/AIDS, psychological problems like anger and frustration, and social issues, which included corrective rape, death, and gender-based violence [35,36,37,38]. The significance of public awareness and transparency about LGBTQI SRHS was indicated as one of the vital approaches to improving SRHS at primary healthcare facilities. This is verified by the studies which detail that the provision of LGBTQI-inclusive care and services should be available [39,40]. The importance of interprofessional communication as a starting point to raise awareness about LGBTQI health matters was also highlighted [41].

Participants recommended training needs about the healthcare services of LGBTQI. This endorses the similar results from these studies [27,39]. A greater need for LGBTQI-specific education to increase the healthcare providers’ competency and comfortability regarding the management and referrals of LGBTQI healthcare [42]. Again, results by Bodeman show that there should be an address to the available deficiency of training opportunities and learning experiences in a couple of areas, such as academia and workplaces, to enable nurses to capably care for LGBT patients [43]. The literature shows that nurses can have the desire to work with and render services to LGBTQI clients through 1-h educational intervention, as a result improving the health of LGBTQI clients [44]. Additionally, participants revealed conflict about the specialization and integration of LGBTQI individuals receiving SRHS. Some stand that there should be specific SRHS for LGBTQI whereas some disagree that if services are specified for LGBTQI individuals only, this might portray discrimination and isolation. Nonetheless, research is in support of the importance of training and education inclusivity that caters to LGBTQI health-related matters [45,46]. In support, the use of simulation has been seen as the most effective method of improving knowledge, skills, and attitudes. This method also addressed the theory-practice gap [47]. Another study indicated that, in the United States, a summit was conducted, which revealed that LGBTQI-specific content in nursing curricula, practice guidelines, faculty development, and research will be necessary to improve the health of LGBQTI people [48].

### 4.1. Limitations of the Study

The study was mapped out to acquire narrative data from PHC nurses who are providing SRHS for LGBTQI individuals. The two major limitations of the study incorporate namely the study population, whereby the study has only been conducted with 27 PHC nurses in Western out of 176 PHC nurses. Also, the study included eight clinics out of 11 clinics in the Western areas. Moreover, the clinics in the other regions of Tshwane, such as Central, Southern, and Eastern clinics and hospitals, were excluded from the study. Lastly, there were limitations associated with the interviews as a method for data collection because some participants were not open about providing answers about LGBTQI issues as they thought they would be reported.

As a result, it is difficult to generalize the study findings as if they are from all PHC nurses in Tshwane.

### 4.2. Recommendations

There should be an incorporation of frequent in-service training, workshops, and continued support to the PHC nurses to assist them in building skills, knowledge, expertise, and an understanding regarding LGBTQI health needs.

Urgent reviewal of the existing and development of new protocols, policies, and guidelines about LGBTQI’s SRHS to be conducted.

LGBTQI talks and campaigns should be implemented by the South African Department of Health to frequently better public engagement, education, and service.

Formulation of the partnership between LGBTQI activists, government, and non-governmental organizations.

The development of LGBTQI health websites to cater to all those who are afraid to access and use SRHS.

South Africa should use simulation, conduct an LGBTQI summit for PHC nurses, and web-based and hours training interventions to assist in addressing theory-practice gaps.

## 5. Conclusions

There is no doubt that according to the PHC nurses’ experiences and perceptions, there are barriers and facilitators for them during the provision of SRHS to LGBTQI individuals. Therefore, expressly, we found that the common barriers for PHC nurses to render SRHS to LGBTQI individuals were related to the institutions, PHC nurses, the general public, and LGBTQI patients themselves. However, regardless of the challenges faced by PHC nurses, there had some facilitators to continue rendering SHRS to LGBTQI patients who came to their clinics.

The facilitators included the obligation of governance and compliance due to their scope of practice, being taught by LGBTQI individuals themselves about their sexual and reproductive practices, and PHC nurses’ work experience and age. Furthermore, almost all PHC nurses suggested the importance of awareness, transparency, specialization, or integration of SRHS and their need for training related to LGBTQI healthcare issues. Future studies regarding similar studies are important to validate the kinds of conclusions that can be drawn from this study. Lastly, if these studies are conducted, knowledge and awareness about LGBTQI individuals’ health issues within healthcare sectors will be increased, thus promoting and preventing all health-risk associated with LGBQTI.

## Figures and Tables

**Table 1 healthcare-10-02208-t001:** Emerged themes and sub-themes.

Themes & Sub-Themes
Barriers to the provision of LGBTQI-related SRHS Sub-themes Institutional-related barriers; PHC nurse-related barriers; Social stigma-related barriers; LGBTQI little openness as a barrier.
Facilitators for the provision of SRHS to LGBTQI individuals Sub-themes The obligation of governance and compliance; LGBTQI individuals as facilitators; PHC nurse’s age; Acquired work experience; Existence of LGBTQI acquaintances, and; The use of mass media.
Strategies to improve LGBTQI individuals’ SRHS accessibility and availability Sub-themes Awareness and transparency of LGBTQI-focused services; Specialization or integration of SRHS; Need for training.

## Data Availability

To ensure the privacy of the participants, raw data is kept under lock and cannot be shared to preserve the anonymity of participants.

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
