# Peer review of "Primary Healthcare Nurse’s Barriers and Facilitators to Providing Sexual and Reproductive Healthcare Services of LGBTQI Individuals: A Qualitative Study"

_healthcare, 2022, doi:10.3390/healthcare10112208_

Round 1
Reviewer 1 Report
General statement:
The study addresses the barriers and facilitators experienced by PHC nurses in providing SRHS to members of the LGBTQ community. Authors display a well-organized overview of current and past literature but need to provide a clearer statement on how this study will add to the existing body of knowledge. The methodology is appropriate for the research but was not appropriately applied and requires revision. Authors did not perform a thematic analysis as indicate in methodology section. Findings contained valuable information but need to be analyzed appropriately to align with research design. Authors need to be aware that some of the sub themes are perpetuating stigma LGBTQI individuals face when accessing health care. The discussion section provided some connection to current and past research in the area but would benefit from a deeper analysis and the integration of the “recommendations” section.
I would like to commend the authors for taking on such an important topic and look forward to seeing it published after extensive revisions.
Abstract:
The abstract includes all necessary sections. The results section did not include the 3 identified themes.
Introduction:
The section was very organized, and the purpose of the study was rooted in the existing literature regarding current circumstances in South African health care for PHC nurses to work effectively with members of the LGBTQI community. Authors need to be clear when citing studies that pertain to the experiences of nurses outside of South Africa. This is not always clear in the first paragraph of the introduction. The abbreviation of LGBTQI also needs to be defined. In line 52-60 authors need to provide more information regarding the challenges they identify. It is not a clear why they equate lack of skills and knowledge with a heteronormative approach. It is a valid point but needs more clarification. Authors identified a gap in research at the end of the introduction. However, paragraphs before state that many studies identify barriers to practice. It might be beneficial to elaborate on what this specific study would add to the existing body of literature.
Methods:
The authors outlined all procedural aspects regarding recruitment, coding, and data analysis. Authors need to be aware of gender inclusive language. Participants should be referred to as “identified as female/male,” not “males and females.”
Regarding the interview guide, no explanation was provided for the difference between general and main questions. The last section of the interview guide is very confusing. Authors refer to refining a tool. It is unclear what type of tool they are referring to. The pre-test and modification procedure needs more detailed information.
Data collection procedure contains some repetitive information from the population and sample size section. Citation for coding framework incomplete. Section covering the data interpretation process is very unclear and needs revising. Section 2.6 ethical considerations are already addressed in the above section.
Results:
The authors did not identify themes but grouped the answers to their specific questions. Each question addressed an area they later identified as a theme which is not an appropriate thematic analysis. The information is very valuable and clearly organized but does not represent accurate qualitative analysis.
Section on LGBTQI-related barriers: This section requires some extensive editing. It is essential to remember who this research study's audience will be. Words are very powerful and, therefore, have the potential to provide support for members of the LGBTQI community or perpetuate existing bias in health care and further alienate this vulnerable population. This current form perpetuates the stigma LGBTQI individuals face.
Section LGBTQI individuals as educators: This section needs revision since it perpetuates the image that health care professionals should utilize their clients as a source for teaching. This has been heavily criticized in the literature regarding LGBTQI-affirming health care practices.
Information provided in table 3 was integrated at the beginning of results section. I would recommend removing table 3 to reduce redundancy of content.
Discussion.
At the beginning of the discussion section, the authors compared some of the results from the present study to findings from other research related to the area of study. However, this section could be strengthened by a more comprehensive analysis of the results. The recommendations section should be included in the discussion and not introduce new terminology such as “queer .”The conclusion section does not outline the potential for future research and societal impact of the topic but provides a list of results that have been mentioned above.
In-text comments:
Line 13: grammar: “the study aims or aimed to..”
Line 18: citation for “Trolley”
Line 19: specific themes need to be included, not just a general comment.
Line 30: word choice: “been achieved well”
Line 37-38: redundancy: not covered in their studies and in the curriculum is referring to the same thing
Line 49 - 51: edit for grammar
Line 70: edit for grammar
Line 86: word choice and grammar: whereas
Line 88: word choice: requisition
Line 93: word choice: open-ended in order
Line 97: unclear. Not sure what is meant by “to refine a tool.”
Line 115-116: grammar.
Line 120: only one person analyzed the data? The manuscript stated: “the researcher”. Please clarify.
Line 120: citation incomplete: Trolley et al (YEAR)
Line 126: word choice: finer meanings
Line 129-131: revise for clarity.
Line 190: edit for repetition “offend”
Line 205-208: revision for grammar and spelling needed
Line 252: word choice. Consider negative connotation of the word enabler
Line 274-275: edit for clarity and content
Line 29- 293: edit for clarity
Line 335-336: word choice: enabler
Line 353: use of abbreviation without explanation
Line 368: citation incomplete
Line 373: word choice: match up
Line 379: word choice: put up
Line 383-384: grammar
Line 385: word choice: enlarge
Author Response
Thank you for the constructive feedback. All corrections were made as advised.
Introduction:
- Authors clarified studies that were not conducted in South Africa in a paragraph e.g. 3,4,5,6 are from overseas. Corrections made.
- The authors defined the LGBTQI acronym as Lesbians, Gays, Bisexuals, Transgender, Queer, and Intersex.
- Clarity was provided in lines 52-60 authors regarding the challenges they identify. The authors shared their observations within South Africa and linked them with incidents that are happening among the LGBTQI in South Africa.
- Clarity provided for the equation of a lack of skills and knowledge with a heteronormative approach.
- Elaboration and clarification have been provided on what this study would add to the existing body of literature.
Methods
- Authors need to be aware of gender-inclusive language. Participants should be referred to as “identified as female/male,” not “males and females.”. Corrected.
- Regarding the interview guide, no explanation was provided for the difference between general and main questions. Clarity provided.
- The last section of the interview guide is very confusing. Authors refer to refining a tool. It is unclear what type of tool they are referring to. The pre-test and modification procedure need more detailed information. Corrections made.
- Data collection procedure contains some repetitive information from the population and sample size section. Corrections made.
- Citation for coding framework incomplete. Corrections made.
- Section covering the data interpretation process is very unclear and needs revising. Corrections made.
- Section 2.6 ethical considerations are already addressed in the above section. Corrections were made to avoid repetition.
Results
The authors did not identify themes but grouped the answers to their specific questions. Each question addressed an area they later identified as a theme which is not an appropriate thematic analysis. The information is very valuable and clearly organized but does not represent accurate qualitative analysis. Corrections were made as advised.
The section on LGBTQI-related barriers: This section requires some extensive editing. It is essential to remember who this research study's audience will be. Words are very powerful and, therefore, have the potential to provide support for members of the LGBTQI community or perpetuate existing bias in health care and further alienate this vulnerable population. This current form perpetuates the stigma LGBTQI individuals face. This section was paraphrased.
Section LGBTQI individuals as educators: This section needs revision since it perpetuates the image that healthcare professionals should utilize their clients as a source for teaching. This has been heavily criticized in the literature regarding LGBTQI-affirming healthcare practices. This section was paraphrased.
Information provided in table 3 was integrated at the beginning of the results section. I would recommend removing table 3 to reduce the redundancy of content. Table 3 removed.
Discussion
At the beginning of the discussion section, the authors compared some of the results from the present study to findings from other research related to the area of study. However, this section could be strengthened by a more comprehensive analysis of the results. A comprehensive analysis of the results was added as advised.
The recommendations section should be included in the discussion and not introduce new terminology such as “queer.”. Corrected to LGBTQI.
Conclusion
The conclusion section does not outline the potential for future research and the societal impact of the topic but provides a list of results that have been mentioned above. Corrections were made accordingly.
In-texts comments:
Line 13: grammar: “the study aims or aimed to..” Corrected.
Line 18: citation for “Trolley” Corrected.
Line 19: specific themes need to be included, not just a general comment. Corrected.
Line 30: word choice: “been achieved well”. Corrected.
Line 37-38: redundancy: not covered in their studies and in the curriculum is referring to the same thing. Corrected.
Line 49 - 51: edit for grammar. Corrected.
Line 70: edit for grammar. Corrected.
Line 86: word choice and grammar: whereas. Corrected.
Line 88: word choice: requisition. Corrected.
Line 93: word choice: open-ended in order. Corrected
Line 97: unclear. Not sure what is meant by “to refine a tool.” Corrected.
Line 115-116: grammar. Corrected.
Line 120: only one person analyzed the data? The manuscript stated: “the researcher”. Please clarify. Corrected.
Line 120: citation incomplete: Trolley et al (YEAR). Corrected.
Line 126: word choice: finer meanings. Corrected.
Line 129-131: revise for clarity. Corrected.
Line 190: edit for repetition “offend”
Line 205-208: revision for grammar and spelling needed. Corrected.
Line 252: word choice. Consider negative connotation of the word enabler. Corrected
Line 274-275: edit for clarity and content. Corrected
Line 291- 293: edit for clarity. Corrected.
Line 335-336: word choice: enabler. Corrected
Line 353: use of abbreviation without explanation. Corrected
Line 368: citation incomplete. Corrected
Line 373: word choice: match up. Corrected
Line 379: word choice: put up. Corrected
Line 383-384: grammar. Corrected.
Line 385: word choice: enlarge. Corrected

Reviewer 2 Report
Dear authors,
Thank you very much for submitting your manuscript “Primary Healthcare Nurse’s barriers and facilitators to providing Sexual and Reproductive Healthcare Services of LGBTQI individuals: A Qualitative Study” to Healthcare
The subject of study is of interest and I believe that it is important to face this type of analysis in all parts of the world to improve the care of this group.
However, the manuscript has a number of shortcomings and the manuscript cannot be accepted in its current form.
The considerations are:
Abstract
The manuscript does not follow the specifications of the instructions for authors. (“The abstract should be a single paragraph and should follow the style of structured abstracts, but without headings.”)
The abstract should not have headings, nor should it include bibliographic citations. In addition, the first time the acronym LGTBTQI appears, the authors must put the full name.
Line 34 SRHS appears for reproductive healthcare services and, later, SRH services. Readers can infer the meaning, but it is not correct to modify the acronyms. In addition, during the text it sometimes appears with the full name and others as an acronym. I recommend the authors to review this aspect.
References
Reference citations do not follow the specifications of the instructions for authors, neither in the text nor in the final section of the manuscript. (In the text, reference numbers should be placed in square brackets [ ], and placed before the punctuation; for example [1], [1–3] or [1,3]. For embedded citations in the text with pagination, use both parentheses and brackets to indicate the reference number and page numbers; for example [5] (p. 10). or [6] (pp. 101–105).)
2. Materials and Methods
Line 70-75 “Data were collected through…” This information does not correspond to 2.1. Type of study. Please review the Material and Method sections to restructure and not repeat information.
The manuscript shows an inconsistency with respect to the sample. In lines 71-72 it states “After interviewing 27 primary health care nurses who were the participants of the study data saturation was reached.” And, later, in line 89-90 it appears “In total there were 23 females and 4 males between the age of 27-63 who participated in the study.” The final sample is 27 because it is the total that agreed to participate? If they only accepted 27, how do the authors show that it is saturated with 27 and not with 28, 29 or 30 participants?
On the other hand, in a qualitative study you must detail the characteristics of the framework or context. The study is taking place in Tshwane, so the authors should provide a description on issues such as location, population, religion, number of people affected by the study, number of nurses in that region, law. In addition, lines 84-86 appear “The researcher explained the purpose and objectives of the study all primary health care nurses during their morning devotion and meetings” This statement shows that the professionals have a specific spiritual context.
Table 1 Does the table show all the questions asked in the interview? The questions shown are few and do not allow knowing the profile of the participant. Qualitative studies should also show who the participants are. Are they experts? What do you know about the subject of study? what previous position do they have? On the other hand, I recommend that you review the wording of the first question and the paragraph that appears after the table. The paragraph is not formatted as text, in fact the line numbers do not appear. This paragraph states that it is asked in English, answered in Setswana, and finally translated into English. What procedure was applied to ensure the correct translation of the interviews? Who or what translated the interview?
2.4. Data collection procedure
The authors must order the content of the section.
The approval of the ethics committee has a specific section and does not correspond to the data collection procedure.
Line 113 Can you add the informed consent that you gave to the participants? A blank sample with no personal data.
Line 120 “…Tolley et al which…” Citation after “et al.” is missing
2.5. Coding and Data Analysis
Who performed the coding and analysis of the data? The content of the paragraph looks like a single person. If so, what expert profile does this researcher have to carry out this process? What techniques did he implement to ensure an objective and replicable process?
2.6. Ethical considerations
Line 138 “Anonymity was maintained throughout data collection and analysis” What actions does the study present to ensure anonymity?
3. Results
The manuscript shows quantity results in letter and number in parentheses. Authors repeat information and may confuse with citations
The tables must appear numbered in order of appearance. Table 3 appears before Table 2 in the text of the manuscript. The style of the tables does not correspond to the standards of the journal.
The title of the table has typos. Ends with “:.” in tables 2 and 3.
The characteristics of the participants are insufficient to describe the sample and I consider that there is a lack of information related to the subject of the study. Why is it interesting to know if they are married or single? Why is no information collected on the opinion of the participants about the LGTBTQI collective? Does any participant have specific training on this topic? Can they modify their speech because they feel studied?
Line 304 What does PrEP mean?
4. Discussion
Line 374-375 Authors should review the writing format because there is a change in font and font size
Line 365 and line 371 End of paragraph references appear in normal size. Authors should review the bibliography.
Line 329-343 This content corresponds to results, not discussion
The authors should strengthen the discussion with more studies that allow their results to be compared.
4.1. Limitations
The limitations of the study can be analysed from different perspectives, not only by the number of participants. In this sense, how many nurses are there in the region under study? This figure would show the partial representation achieved.
The authors must include analysis of limitation in the methodology, from the questionnaire to the procedure.
4.2. Recommendations
Line 415-414 What do the authors mean by “Same research should be conducted using different media, for example, films and podcasts, and there should be involvement of both, nurses, and the LGBTQI community”?
5. Conclusions
The conclusions must return to the objective and not only summarize the main findings that have been previously exposed in a similar way.
Author Response
Thank you for the constructive feedback. We appreciate it, please note that we have worked on the corrections as advised.
Abstract:
- Headings and bibliographic citations from the abstract are removed.
- The authors defined the LGBTQI acronym as Lesbians, Gays, Bisexuals, Transgender, Queer, and Intersex as it was appearing for the first time.
Introduction:
Line 35 SRH service corrected to SRHS for consistency with line 34 as observed by reviewers 2 & 3.
Methods:
- Line 70-75 of “Data were collected through, was changed and moved to a procedure and paraphrased to “An exploratory approach guided by a semi-structured interview guide and face-to-face interviews was used”.
- Consistency maintained by writing “Data saturation at 24th interview and the researcher stopped collecting data on the 27th interview”
- Line 89-90 corrected for consistency
-
- Line 84-86, not all the sites had morning devotions, again. In South Africa, it is common that most of the clinic/health care facilities before starting work, they prayer nurses pray regardless of the spiritual context.
- The authors gave an overview description of Tshwane, and its population, indicating that all members of LGBTQI, their families, and significant others are affected (No specific statistics as they have little knowledge about the study, however, due to work experiences and challenges such as murders, homophobic observed around the region), the religion of Tshwane communities, number of nurses in Tshwane West, law by the country and National Department of Health.
- Line 84-86, not all the sites had morning devotions, again. In South Africa, it is common that most of the clinic/health care facilities before starting work, pray for nurses pray regardless of the spiritual context.
- Corrections made in Table one Table 1: All questions (Demographic, general, main, and probing) are included. This briefly provided information about who the participants are, whether are they experts, and what they know about the subject in Table 2.
I.e. Demographic questions:
Place of data collection, age, gender, marital status, rank & specialty, and how long have you been a nurse?
General questions:
- What type of services do you render in your institution as prescribed by the Department of Health?
Main questions:
- Main: Could you please share your experiences when rendering SRHS to LGBTQI?
Probe:
- How do you feel about the current services offered to them?
- How was your interaction with LGBTQI patients?
- Main: What utmost challenges have you experienced when you have to render SRHS to the LGBTQI individuals in your clinic?
Probe:
- What makes it difficult?
- How prepared/comfortable are you working with LGBTQI patients? (Condom use, profoundly oral sex, anal sex, sex toys).
- Main: What keeps on encouraging you to continue providing SRHS to LGBTQI individuals, in spite of the challenges?
Probe:
- How did/do you learn to provide sexual and reproductive health care services to LGBTQI individuals? (type of training received).
- Main: What can you suggest so that SRHS for LGBTQI individuals can be improved?
Probe:
- What other suggestions do you have in mind that you think can help improve sexual and reproductive health care services for LGBTQI individuals?
- This paragraph states that it is asked in English, answered in Setswana, and finally translated into English. What procedure was applied to ensure the correct translation of the interviews? Who or what translated the interview? Clarity has been given in 2.5 under data analysis. However, they indicated that readers should refer to section 2.
-
2.4. Data collection procedure
The authors must order the content of the section. The approval of the ethics committee has a specific section and does not correspond to the data collection procedure: Was removed as also advised by reviewer 3, and it was moved to the ethical considerations
Line 113 Can you add the informed consent that you gave to the participants? A blank sample with no personal data: A sample of a blank consent form was added as an annexure after references.
Line 120 “…Tolley et al which…” Citation after “et al.” is missing: A year and number28
-
2.5. Coding and Data Analysis
Who performed the coding and analysis of the data? The content of the paragraph looks like a single person. If so, what expert profile does this researcher have to carry out this process? What techniques did he implement to ensure an objective and replicable process? : Correction made, The process was led by one researcher whereby it was conducted after receiving training from the university, the researcher handed over the analyzed data to the supervisor as a peer reviewer to ensure objectivity and replication.
2.6. Ethical considerations
Line 138 “Anonymity was maintained throughout data collection and analysis” What actions does the study present to ensure anonymity? Participants were requested not to mention their names, each participant was given a false name such as P1 which stands for participant 1, and each had a number. The researcher addressed them as P and added a number.
-
The manuscript shows quantity results in letters and numbers in parentheses. : This is to show the duration of service per participate, (YOS stands for Years of Service, GN, General nursing, PHC, Primary Health Care Nursing, and ONC, Occupational Nursing).
Authors repeat information and may confuse with citations: Clarity given above and corrections made.
The tables must appear numbered in order of appearance. Table 3 appears before Table 2 in the text of the manuscript. The style of the tables does not correspond to the standards of the journal. Tables were corrected, and Table three was removed as requested and suggested by other reviewers. Corrected on Table 1 which was 2 corrected.
The title of the table has typos. Ends with “:.” in tables 2 and 3. Corrected.
The characteristics of the participants are insufficient to describe the sample and I consider that there is a lack of information related to the subject of the study. Why is it interesting to know if they are married or single? Why is no information collected on the opinion of the participants about the LGTBTQI collective? Does any participant have specific training on this topic? Can they modify their speech because they feel studied?: As indicated, table 3 for the characteristics was removed as advised and suggested by reviewers 1 & 3. Information of questions or probes had their perception of LGBTQI SRHS, It was added as advised. As indicated above they have different specialties, however, less according to the curriculum of South Africa they are not taught about LGBTQI but are expected to render services for them
Line 304 What does PrEP mean? Corrected and written in Full (Pre-exposure prophylaxis)
Discussion:
Reviewer 2 corrections made:
Line 374-375 Authors should review the writing format because there is a change in font and font size: Changed and corrected
Line 365 and line 371 End of paragraph references appear in normal size. Authors should review the bibliography: Changed and corrected
Line 329-343 This content corresponds to results, not the discussion. This is introductory for the discussion, taking the reader from results and introducing discussions. The authors continued discussing it.
The authors should strengthen the discussion with more studies that allow their results to be compared. Corrections done and more study added as advised.
4.1. Limitations
The limitations of the study can be analyzed from different perspectives, not only by the number of participants. In this sense, how many nurses are there in the region under study? This figure would show the partial representation achieved. The authors must include an analysis of limitations in the methodology, from the questionnaire to the procedure. Corrected as advised.
4.2. Recommendations
Line 415-414 What do the authors mean by “Same research should be conducted using different media, for example, films and podcasts, and there should be involvement of both, nurses, and the LGBTQI community”? Corrected and new recommendations added due to the discussions.
Conclusion:
The conclusions must return to the objective and not only summarize the main findings that have been previously exposed in a similar way. Corrections done.

Reviewer 3 Report
The manuscript entitled "Primary healthcare nurse's barriers and facilitators to providing sexual and reproductive healthcare services of individual LGBTQI: A qualitative study" aims to explore the perception of primary healthcare nurses regarding the care they provide to LGBTQI individuals, within the scope of sexual and reproductive health, conducting semi-structured interviews with 27 nurses.
The manuscript has more than 80% of the references are less than 5 years old, covering documents from the World Health Organization, publications from indexed journals with a high impact factor and theses in the area of ​​study.
The manuscript is structured according to the publication guidelines of the Healthcare journal.
I want to congratulate the authors for their interest in this thematic area, above all for wanting to improve the care provided to the LGBTIQ community, often discriminated against by society.
After carefully reading the manuscript, I leave some suggestions and reflections in order to improve the manuscript.
Regarding affiliation information:
> Both authors have the same affiliation, so they could change the numbering from "Mokgatle MM, PhD" to 1 (line 5) and in the affiliations just leave the number "1 Sefako Makgatho Health Sciences University". Correct line 8 so as to separate the email name.
Regarding the summary:
> Give a space between "Tolley." and "Results:" (line 18).
> The keywords "influencing" and "factors" are very vague and are not MeSH. I suggest deleting them.
> The keyword "Primary healthcare nurses" is also not MeSH, so I suggest changing it to "Primary health care" and "Nurses" or "Nursing".
> It can be advantageous to have "Qualitative study" as a keyword.
Regarding the introduction:
> On line 35 replace "SRH services" -> "SRHS".
Regarding the type of study:
> The information starting on line 70 and ending on line 75 ("Data were collected ... analyze them" wouldn't look better in the "Data collection procedure" section?
Regarding the interview guide:
> Given that changes were made to the interview after the pre-test was carried out, is it correct to use the pre-test data in the data analysis?
> Information presented as "Table 1" would be better presented as a list of questions rather than a table.
Regarding the data collection procedure:
> Wouldn't the information starting on line 104 and ending on line 108 ("The research and Ethics ... Tshwane municipal clinics") look better in the ethical considerations section?
Regarding the results:
> On line 147 change "sub-themes:." -> "sub-themes.".
> Tables must not have vertical lines (Table 2)
> Is table 3 really necessary to characterize the participants? Why not a descriptive paragraph? If you keep Table 2, it should not have vertical lines and the abbreviation "YOF" is not specified.
> The introductory paragraph "Theme 1: Barriers to the provision of LGBTQI-related SRHS" (lines 145–156) is round, so I would suggest removing it.
> In Table 2 the sub-themes have slightly different nomenclature, which suggests uniformity throughout the manuscript (for example: with or without -related no in the sub-themes of theme 1; obligation of governance and compliance vs governance and compliance; younger age of the PHC nurses as an enabler vs PHC nurse's work experience and age...).
> The participant quote referenced in lines 177–181 is not related to the sub-theme “PHC nurses-related barriers”?
> The description of the sub-theme "Public-related barriers" makes me think that it would be more understandable if the sub-theme was called "Social stigma-related barriers".
> In the sub-theme "LGBTQI-related barriers" don't you consider that the terms "dissimulation" or "dishonesty" have a negative connotation in relation to LGBTQI? It is different to have little openness to talk about sexuality and to be dishonest or underhanded. I would suggest changing the sub-theme description and even the sub-theme name.
In the discussion section:
> In line 337 change "of theme 2 are;" -> "of theme 2 are:", and on line 243 change "sub-themes;" -> "sub-themes:".
> On line 253, "SA" appears for the first time. I believe it means South Africa.
> Line 363: Replace "LGBQTI" -> "LGBTQI".
> Place references 32 (line 365) and 33 (line 371) in the reference format (32 and 33).
> Standardize the font on lines 374–375 ("the negative consequences of lack of the SRHS to").
In the study limitations section:
> On line 399 replace "namely;" -> "namely:"
In the references section:
> Reference 41 does not specify the year.
Author Response
Thank you for the constructive feedback. We have corrected all the suggestions and corrections.
Affiliation Information:
- Affiliation corrected. The authors changed the numbering of Mokgatle MM, Ph.D. to 1.
- Affiliation number 2 was deleted to leave number 1 as both authors belong to the same affiliation.
- Line 8 was corrected; name and email were separated.
Abstract:
- Influencing and factors as keywords were removed from the abstract.
- The authors added a Qualitative study as a new keyword as advised by the reviewers.
- Other new keywords as advised are, Primary healthcare, and Nurses.
Introduction:
Line 35 SRH service corrected to SRHS for consistency with line 34 as observed by reviewers 2 & 3.
Methods:
Reviewer 3 corrections made:
Regarding the type of study:
- Sentence starting with Data collected moved to Procedure section
Regarding the interview guide:
- The results for pre-test are important as they form part of the study. They were few changes made in only how questions should be phrased.
- Information in Table 1 was changed to “List of questions” All Tables were corrected.
Regarding the data collection procedure:
The sentence starting with Ethics moved to ethical considerations.
Results:
Reviewer 3 corrections made:
- Line 147 corrected
- Vertical lines removed on Table 2
- Table 3 was removed as advised by reviewers 1 & 3
- Introductory paragraph for Theme 1 was removed as requested
- All themes and sub-themes corrected. There is consistency and similarities in the table and discussions.
- Quotations under PHC nurses correct and added new ones that are related to PHC-related barriers
- Sub-theme for Public-related barriers changed to Social stigma-related barriers
LGBTQI-related them clarified, corrected to “LGBTQI little openness as a barrier” and corrected the paragraph to avoid being/sounding negative towards LGBTQI individuals as advised.
Discussion:
Reviewer 3 corrections made:
- It is not clear what the reviewer is asking the authors to change, as the requested change looks the same. Therefore, the authors changed it to “Theme 2’ sub-themes are” to make it clearer.
- SA corrected to South Africa in full.
- All the inconsistencies for LGBQTI have been corrected to LGBTQI
- Correct referencing for 32 and 33 have been made to 32 &33
- Font standardized accordingly
References:
Reviewer 3 corrections made:
- Reference 41, year added.

Round 2
Reviewer 1 Report
I appreciate the responsiveness to my comments. It is a vast improvement, and I look forward to seeing it published.